# Ultrasound-Assisted Extraction of Cannabinoids from Cannabis Sativa for Medicinal Purpose

**DOI:** 10.3390/pharmaceutics14122718

**Published:** 2022-12-05

**Authors:** Antonella Casiraghi, Andrea Gentile, Francesca Selmin, Chiara Grazia Milena Gennari, Eleonora Casagni, Gabriella Roda, Gloria Pallotti, Pierangela Rovellini, Paola Minghetti

**Affiliations:** 1Department of Pharmaceutical Sciences, Università degli Studi di Milano, Via Giuseppe Colombo 71, 20133 Milano, Italy; 2INNOVHUB-Stazioni Sperimentali per l’Industria S.r.l., Area Oli e Grassi, Via Giuseppe Colombo 79, 20133 Milano, Italy

**Keywords:** sonotrode, FM2, Bedrocan, oil characterization, DSC, lipophilic volatile carbonyl compounds, oxidized fatty acids, tocopherol

## Abstract

Over the past 20 years, the interest in Cannabis oily extracts for medicinal use compounded in pharmacy has consistently grown, along with the need to have preparations of adequate quality. Hot maceration (M) is the most frequently used method to compound oily solutions. In this work, we systematically studied the possibility of using an ultrasonic homogenizer and a sonotrode (US) as an alternative extraction method. Oily solutions were prepared using two available varieties of Cannabis for medicinal use, called FM2 and Bedrocan. All preparations resulted with an equivalent content in CBD and THC, with the advantage of a faster process using US. In particular, 10 min sonication at the amplitude optimized for the sonotrode used (2 or 7 mm) provides not statistically different total Δ9-tetrahydrocannabinol (M-FM2: 0.26 ± 0.02 % w/w; US-FM2: 0.19 ± 0.004 % w/w; M-Bedrocan: 1.83 ± 0.17 % w/w; US-Bedrocan: 1.98 ± 0.01 % w/w) and total cannabidiol (M-FM2: 0.59 ± 0.04 % w/w; US-FM2: 0.58 ± 0.01 % w/w) amounts extracted in refined olive oil. It can therefore be confirmed that sonotrode is an efficient and fast extraction technique and its use is without negative consequence on the solvent properties. Despite DSC evidencing that both maceration and sonication modify the T_onset_ and enthalpy of the event at about −10 °C, the qualitative characteristics of the oil remained constant for the two treatments and similar to the starting material.

## 1. Introduction

The use of Cannabis sativa has recently received more attention: its main components, Δ9-tetrahydrocannabinol (Δ9-THC), with psychotropic activity, and cannabidiol (CBD), non-psychoactive [1], can be used as symptomatic supportive treatment in many diseases, i.e., to reduce convulsions in cases of childhood and/or adolescent epilepsy [2], to manage neuropathic pain [3] and counteract the side effects of chemotherapy [4]. Other than two main components, up to 125 other structurally related compounds, called cannabinoids, [5] have been extracted and characterized to date [6]. Since the synergies existing in the phytocomplex can increase the activity of Δ9-THC and limit side effects, medicinal products should be based on the natural chemical complexity of Cannabis, rather than using a single [7] or synthetic cannabinoid [8]. As the availability of authorised medicinal products is limited [9], compounding activity in pharmacy is the way to guarantee these treatments. However, pharmacists need to take into consideration two aspects strictly connected: conversion to the active form and the extraction of cannabinoids. Indeed, the natural carboxylated form, namely Δ9-tetrahydrocannabinolic acid (THCA) and cannabidiolic acid (CBDA), require a decarboxylation process to be converted in the neutral and more pharmacologically active form (Δ9-THC and CBD, respectively). This conversion is catalyzed by light, aging, and exposure to heat [10,11,12]. In the case of heating, method used when oily solutions are compounded in pharmacies, temperature and the time of exposure dictate the degree of decarboxylation, without causing the degradation of the cannabinoids [13]. In fact, Δ9-THC can react with atmospheric oxygen to give various degradation products. It is also worth noting that decarboxylation of a plant matrix is not a strictly stoichiometric reaction, since the reaction conditions and the intrinsic reactivity of the single cannabinoids influence the degradation or reaction with other components [11,14]. Finally, heat-induced alterations of the extraction solvent should also be avoided. 

Since cannabinoids are lipophilic substances, they are more likely to dissolve in non-polar solvents, such as olive oil or medium chain triglycerides. Extraction can also be performed with various polar and non-polar organic solvents, among them supercritical CO_2_ [15] or ethanol [16]. However, the use of ethanol is questionable. First, it can solubilize chlorophyll responsible of the unpleasant taste. Then, the uncomplete evaporation results in a very viscous semifluid mass from which it is difficult to remove the last residues. At the same time, prolonged heating causes the loss of the terpenic components and potential degradation of Δ9-THC into cannabinol (CBN). As a matter of fact, the degradation of main cannabinoids was greater in ethanol than in olive oil after 10 days of storage at the temperature of 8 °C: the Δ9-THC content decreases by about 1.5% and 22.5% in olive oil and in ethanol, respectively; while at 25 °C the loss is about 44% and 82% in oil and alcohol, respectively [17]. Finally, residues in the extract can cause high irritation for the mucous membranes of the gastrointestinal tract [17]. 

Among edible oils, olive oil is excellent to effectively retain the terpene fraction of Cannabis upon heating [18]. The extraction process in olive oil can be based on maceration and ultrasonic assisted extraction (UAE). The latter is gaining increasing attention as an alternative since UAE is efficient in terms of time and final yield [19,20]. UAE is suggested to overcome some drawbacks of other extraction techniques. The effect of cavitation energy leads to the destruction of plant cells and, thus, improves the release of intracellular substances in the solvent [21]. UAE can be carried out using two types of ultrasound equipment: (a) an ultrasonic water bath, or (b) an ultrasonic probe system fitted to horn transducers, the sonotrode. This is reported to be an efficient extraction technique because the direct contact with the sample allows the development of a power up to 100 times more than that provided in the ultrasonic bath, which leads to shorter extraction times when compared to the ultrasonic bath [22]. As ultrasound-catalyzed reactions and degradation of the extracted molecules due to free radicals formed in the extraction solvent may occur, an extensive study of extraction parameters (i.e., temperature, the dielectric constant of the solvent, power, intensity, frequency and the time of application of the ultrasounds) is required [23,24,25,26] and also in the function of the equipment used [16,27]. In the case of extraction of cannabinoids, previously reported results obtained using UAE were performed only using an ultrasonic bath [27,28,29,30]. 

The aim of the present work was to optimize the preparation of Cannabis oily extracts for medicinal use, using the UAE technique, in order to be able to achieve the same results obtained with the methods already in use in pharmacies, basically based on the maceration extraction. 

## 2. Materials and Methods

### 2.1. Materials

FM2, produced by Stabilimento Chimico Farmaceutico Militare (Firenze, Italy). It is available in dried and ground form. It is declared to have a balanced ratio of THC 5–8% and CBD 7.5–12%. A batch for scientific use with a THC and CBD content out of specification was used. Bedrocan^®^, brand name for the cultivar *Cannabis sativa* L. ‘Afina’ (Bedrocan International, Amsterdam, The Netherlands) is available in flos form. It features 22% THC, with a CBD-level below 1%. Refined olive oil Ph. Eur. was obtained from Societe Industrielle des Oleagineaux (Saint-Laurent-Blangy, France). It was vacuum packed. Fatty acid composition of used batch was 9.70% palmitic acid, 0.60% palmitoleic acid, 3.20% stearic acid, 80.10% oleic acid, 4.70% linoleic acid, 0.60% linolenic acid, 0.40% arachidic acid, 0.30% eicosenoic acid, <0.10% behenic acid, and <0.10% lignoceric acid. 

To obtain the decarboxylation of the acidic cannabinoids naturally present in Cannabis, plant material was heated in a petri dish put in an oven (Memmert UN30, Memmert, Büchenbach, Germany) at 115 °C for 40 min. FM2 samples were also prepared at 125 °C for 40 min, to evaluate the temperature effect. Unheated samples were used as control. Afterward, the extraction was carried out by maceration or by ultrasonic probe. 

### 2.2. Chemicals, Reagents and HPLC Solvents

Cannibinoid analysis: Methanol, acetonitrile, phosphoric acid, (-)-D9-THC methanol solution at 1 mg/mL, CBD methanol solution at 1 mg/mL, standard solutions at 1 mg/mL in acetonitrile of THCA, CBDA, CBN (all analytical grade > 99%) were purchased from Sigma-Aldrich (St. Louis, MI, USA). Water (18.2 Ω cm^−1^) was prepared using a Milli-Q System (Millipore, Darmstadt, Germany). Olive oil analysis: 2,4-dinitrophenylhydrazine (reagent grade, 97%) and sodium benzyloxide solution (0.1 M solution benzyl alcohol) were purchased from Merck, Darmstadt, Germany; quantification and recovery standards were purchased from Sigma-Aldrich (St. Louis, MI, USA), (±)-α-Tocoferol (≥96% HPLC grade), D-α-tocopherylquinone (≥96% HPLC grade), lauric aldehyde (purity ≥ 95%); tricaproin and triheptadecanoin (purity ≥ 99%) from Larodan, Solna, Sweden.

### 2.3. Cannabis Extract in Olive Oil

*Maceration*: 2 g of finely grinded Cannabis, decarboxylated or not, were added to 20 mL of olive oil. A mixer was used to further crumble the plant material. Then, the open beaker was put in a silicone-oil bath; 350 mL solvent was pre-heated at fixed temperature (100 °C) and maintained in constant stirring. The mixture was stirred for 40 min and then immediately filtered to obtain the final oil. Preparation was also performed with 5 g of Cannabis and 50 mL of olive oil [13] in order to reproduce in-use pharmacist conditions.

*Ultrasound assisted extraction:* the extraction was conducted using an ultrasonic system equipped with sonotrode (UP200St, 200 W–26 kHz, Hielscher, Teltow, Germany). An amount of 2 g of finely grinded Cannabis, decarboxylated or not, were dispersed in 20 mL of olive oil at room temperature (25–27 °C) and the extraction was conducted with 2 mm sonotrode (S26d). For 20 mL sample different sonication times were used (10, 20 and 30 min) with an amplitude of 60%. The solution was filtered to obtain the final product. The same preparation was also performed increasing the solvent volume but maintaining the same ratio for grinded Cannabis and olive oil (i.e., 5 g of sample in 50 mL of olive oil). As suggested by the manufacturer for volume up to 20 mL, a 7 mm sonotrode (S26d7) was used and samples were sonicated for 10 min using an amplitude of 30 or 35%. All conditions are summarized in Table 1.

### 2.4. Quantitative Determination of Cannabinoids

The analytical system consisted in an HPLC/UV Prominence-i LC-2030C-Cannabis Analyzer for Potency (Shimadzu Corporation, Kyoto, Japan). Separation was attained on a reversed-phase Shimadzu NexLeaf CBX for Potency, 2.7 μm (150 mm × 4.6 mm) analytical column, preceded by a security guard cartridge. The linear gradient was between eluent A (water) and eluent B (acetonitrile) both containing 0.085% phosphoric acid. The flow rate was 1.6 mL/min and the column temperature was 35 °C. The elution gradient was set as follows: 0–3.0 min (70% B), 3.1–7.0 min (70–85% B), 7.0–7.1 min (85–95% B), 7.1–8.0 min (95% B), 8.0–8.1 min (95–70% B) and 8.1–10 min (70% B). UV detection was monitored at 220 nm. Qualitative analyses were performed on 11 cannabinoids; quantification was restricted to 5 cannabinoids: CBDA, CBD, CBN, Δ9–THC, THCA (LOD 0.002%, LOQ 0.005%).

### 2.5. Olive Oil Characterization

Three independent samples of refined olive oil, untreated and after the two extraction methods were analyzed in terms of carbonylic volatile organic compounds (CVCs), aldehydes and ketones originated from autoxidation of fatty acids; oxidized and conjugated fatty acids; and tocopherol and their oxidized forms. DSC analysis was also performed.

According to our internal method [31], CVCs were derivatized with a 2,4-dinitrophenylhydrazine solution (0.1% in acetonitrile 0.01N HClO_4_) and chromatographically separated, identified, and quantified using HPLC-PDA (P400 with UV6000 LP, Thermo Finningan, San Jose, MA, USA). HPLC coupled with UV with spectra acquisition range from 200 to 400 nm was used. Aliquots of 0.10 ± 0.01 g oil were placed in test tube with 50 µL of lauric aldehyde solution (500 mg/L) as internal standard. After the addition of 1 mL derivative solution, the reaction is promoted by ultrasonic bath for 15 min at room temperature. The chromatographic separation was obtained with a ternary linear gradient of 60 min from water:acetonitrile:methanol 40:30:30 to 0:50:50 v:v:v, at flow rate of 0.9 mL/min. The quantification was performed at 360 nm, using lauric aldehyde as internal standard.

Oxidized and conjugated fatty acids were determined as benzyl derivates [32]. HPLC Nexera equipped with SIL-20A (Shimadzu Corporation, Kyoto, Japan) coupled with UV (SPD 20A UV, Shimadzu Corporation, Kyoto, Japan) with spectra acquisition range from 200 to 400 nm. The method is based on the transesterification of fatty acids with a sodium benzyloxide solution at 1.0 M. About 0.50 ± 0.01 g of sample was put in 25 mL volumetric flask and made to volume with hexane. Afterwards, 1 mL of solution was transferred in a test tube with 1 mL of mix solution of tricaproin (200 mg/L) and triheptadecanoin (400 mg/L). The sample was allowed to react 15 min after addition of 50 µL benzyloxide solution. After centrifugation, the upper layer was dried under nitrogen at room temperature; 20 µL of reconstituted sample with 2-propanol were injected in the HPLC with a photo diode array (PDA) detector. An HPLC gradient was applied from 40:60 to 0:100 v:v water:acetonitrile in 50 min at the flow rate 1 mL/min. The chromatograms were obtained at 255 nm. All compounds were quantified as triheptadecanoin; complete transesterification control takes place by calculating the ratio between the response factors (RF) of tricaproin and triheptadecanoin.

For the determination of tocopherol and their oxidized forms, about 0.30 ± 0.01 g of homogenized oil was placed in a 10 mL amber volumetric flask, made up to volume with acetone and vigorously shaken. An HPLC (P400 with UV6000 LP, Thermo Finningan, Massachusetts, USA) coupled with UV with spectra acquisition range from 200 to 400 nm was used. The chromatographic separation was performed on Luna C18 (250 × 4.6 mm, 5 µm (Phenomenex, Torrance, CA, USA); mobile phase: 40:30:30 v:v:v water:methanol:acetonitrile, maintained for 15 min at the flow rate 1.5 mL/min; after 25 min acetonitrile and methanol increase to 50:50 v:v. The chromatograms were obtained at 292 nm for tocopherol and 268 nm for oxidized forms.

Finally, Differential Scanning Calorimetry (DSC) was carried out on olive oils using a DSC 1 Stare system (Mettler Toledo, Greifensee, Switzerland) operating with a Stare software. Samples of about 15 µL accurately weighted, were sealed in 40-µL aluminum pans and subject to two cooling and heating cycles from 25 to −60 °C at the cooling and heating rate of 5 K/min. DSC cell was purged with a dry nitrogen flow of 80 mL/min. The system was calibrated using an indium standard. Data were treated with Stare System software (Mettler Toledo, Switzerland).

### 2.6. Statistical Analysis

Data are reported as the means ± standard deviation. Comparisons among the groups were performed by analysis of variance with Bonferroni correction. *p* < 0.05 was considered significant.

## 3. Results and Discussion

### 3.1. Decarboxylation of Cannabis FM2

In this study, Cannabis FM2 for scientific purpose was used, available in a dried and grounded form, in a container of 5 g each. The grinding process carried out by the manufacturer allows to eliminate the variability due to the composition of a single inflorescence and the presence of the central rachis (i.e., central branch to the inflorescence), but it could also cause a partial separation of glandular trichomes (i.e., the structures in which cannabinoids are synthesized and accumulated) in the container, leading to different cannabinoids content in the sample if it is used only partially. To set up the UAE conditions, samples were prepared using 40% of the total amount present in the container and hence the plant material was manually mixed before sample preparation to avoid a lack of uniformity. The variability in cannabinoid content in the raw material was within the value of 8% (Table 1). Limited amounts of neutral forms and CBN, namely the aromatization product of Δ9-THC, were also detected. It is worth mentioning that CBN is considered as a reference marker for oxidative degradation and, therefore, used as an indicator of the quality of Cannabis [33]. The amount present in the raw materials was probably a consequence of a long period of storage from production to use. 

The effect of temperature on the plant material was measured at 115 °C [13] and 125 °C. Higher temperatures were not considered, as from 130 °C increasing amounts of CBN were measured [13]. The increase of temperature influenced the degree of CBDA conversion (Table 1). Even if the obtained values were not statistically different, the higher temperature was more suitable for the decarboxylation of Cannabis chemotypes with similar CBD and THC content, i.e., FM2. The conversion of CBDA into CBD at 125 °C was almost complete (over 96%), while at 115 °C the neutral form remained below 90%. On the contrary, detectable residues of Δ9-THCA remained only at 115 °C (Table 1). This was due to the higher activation energy required by CBDA compared to Δ9-THCA to obtain the decarboxylation reaction and it is in line with literature data since decarboxylation close to 130 °C led to a greater conversion of CBDA [11]. Similarly, both temperatures doubled the CBN content with respect to those quantified in FM2 (Table 2). 

### 3.2. Cannabinoid Content in Samples Prepared by Maceration Method

The impact of the decarboxylation process on the extraction of cannabinoids from plant material subjected at the two different decarboxylation temperatures was evaluated and compared with results obtained using the untreated FM2 (Table 2, M-FM2). A quite great variability was noted for almost all extracted cannabinoids. As expected without preheating, the extraction of cannabinoids in acidic form was prevalent, while after decarboxylation lower values were measured at 115 °C and their complete absence was observed at 125 °C (Table 3). The different effect of the decarboxylation temperature is highlighted by the presence of CBDA quantifiable amounts only at a temperature of 115 °C. With respect to cannabinoids content, only total CBD, samples M-FM2 vs M-FM2_115, was significantly different (Bonferroni-Holm test, *p* = 0.02). 

The preparation of oily extracts according to the maceration method [13] requires an extraction time of 40 min, at a temperature of 100 °C. The heating medium is represented by silicone oil, which is warmed up to the required temperature before immersing the beaker containing Cannabis dispersed in olive oil. To check this process, the variation of silicone bath temperature was monitored over time (Figure 1). Under our experimental conditions, as soon as the 50 mL-sample was immersed in the heating bath, a drop in silicone oil temperature of about 10 degrees was observed due to the subtraction of thermal energy by the sample and it took 10 min to return to fixed value. On the other hand, the same time was necessary for olive oil to reach a constant temperature. Then, the extraction proceeded in isothermal conditions. Although the magnitude of these events is related to extraction equipment, temperature variations have an overall impact on the duration of the process. Only high temperatures reduce the viscosity of the oil, improving its diffusivity in the cells of plant material and the extraction yield solvent [21].

### 3.3. Cannabinoid Content in Samples Prepared by Ultrasound-Assisted Extraction

The preparation of the samples was carried out maintaining the Cannabis/olive oil ratio 1/10, as already optimized in a previous work [13]. Results on 20 mL of olive oil samples containing untreated FM2 were similar to those obtained with samples prepared by maceration (Table 3, M-FM2), highlighting that the UAE does not catalyse the decarboxylation reactions or conversion into the neutral forms (20 min exposure). As already observed after maceration, a slight increase in CBN was observed, probably due to the oxidation of Δ9-THC (Table 4). Total CBD and total THC content were similar and comparable between the two methods.

The results on FM2 composition, which was decarboxylated at 115 °C and sonicated (amplitude 60%) in 20 mL are reported in Table 4. As expected, the respective acid forms of Δ9-THC and CBD were strongly reduced or completely absent. It is evident that the same percentage extraction efficiency was obtained at each extraction time considered, namely 10, 20 and 30 min. 

Gajic et al. [21] reported similar results optimizing UAE using a central composite design for extraction of carotenoids from orange peel with olive oil. According to the authors, in the first phase of extraction, the desired compounds were leached from the surface of the damaged cell walls. The longer effect of cavitation energy damaged the cell membranes, which led to the release of carotenoids.

Furthermore, comparing the results obtained with maceration (Table 2, M-FM2_115) and ultrasound (Table 4), CBD and THC concentrations, in their neutral form and as total values, were not significantly different. This means that with a sonication frequency of 26 kHz and amplitude 60%, applied for 10 min, cannabinoid extraction is comparable to that obtained by maceration after 40 min at 100 °C. The sonication method had only a slightly higher effect on the degradation of Δ9-THC; the average content of CBN was statistically different between the two method (*p* value < 0.05). 

To assess the robustness of this protocol, solvent volume was increased to 50 mL and both the 2 mm and 7 mm sonotrode were used; the latter is suggested by the manufacturer to be used with a larger volume of solvent. Different geometries and sizes of the sonotrode lead to a different efficiency in transmitted energy, which is related to amplitude. Using 2 mm sonotrode and 60% amplitude, in 20 mL samples, the measured energy density at 10 min was 228 ± 22 J/mL. Using the same parameters in 50 mL oil, energy density decreased to 119 ± 3 J/mL (Figure 2). To adjust the power applied during the 2 mm sonotrode treatment to that of 7 mm sonotrode, amplitude was reduced at 30%; measured energy density was 227 ± 5 J/mL.

The efficiency of UAE against maceration method was evaluated comparing total THC and total CBD extracted with 2 mm sonotrode in 50 mL (Table 5) with the same samples (Table 3, M-FM2_115); they are slightly lower for UAE. However, no statistical differences were calculated both for THC and CBD (*p* > 0.05). 

Using the 7 mm sonotrode a minor variability of extracted cannabinoids was observed (Holm-Sidak test, *p* = 0.02). The surprisingly low THC content was probably due to the variability of raw material.

### 3.4. Preparation of Bedrocan Samples by UAE

Bedrocan is one of the most prescribed Cannabis chemotypes, therefore UAE was tested for preparation of samples required by physician prescriptions. 

Conditions previously fixed for FM2 were used. Decarboxylation was performed at 115 °C. Results obtained after maceration were compared to those obtained after UAE applied for 10 min at different amplitude (Table 6). THC and total THC extracted after maceration or UAE were not significantly different (*p* > 0.05). CBN was always below 0.05% and the high variability of THCA could be due to very low detected amounts.

The effect of different amplitude was evaluated. At 35% amplitude the measured energy density was 253 ± 6 J/mL. Over 10 min ultrasound application, the temperature increases continuously (Figure 2), even if milder values are reached compared to maceration. The increase in amplitude led to an increase in the cavitation effects, which provides the formation and collapse of the cavitation bubbles during wave propagation. The implosion of the bubbles generates microjets and solvent flows, which in turn led to the cell rupture and mass transfer increasing the release of compounds from the matrix into the solvent [34]. Even if slightly different amplitude was used, the trend of this effect was observed (Table 6).

### 3.5. Olive Oil Components and Effect of the Extraction Procedures

To prove the success in cannabinoid extraction by reducing the time of preparation to 10 min, changes in the olive oil due the extraction processes were evaluated, monitoring the composition, lipophilic volatile carbonyl compounds, the content of tocopherols and oxidized tocopherols and the oxidation state of the oil components. 

The Carbonylic Volatile Compounds (CVCs) are mainly responsible for the sensory notes of the flavor, in particular the ketone-aldehyde fraction contributes for the most part to the natural flavor of olive oil and can undergo auto-oxidation or photo-oxidation processes. The formation of C6 and C9 aldehydes starting from linolenic and linoleic polyunsaturated fatty acids by breakdown of hydroperoxides according to self-oxidation or retro-aldo degradation mechanisms was monitored by means of HPLC analysis and therefore can be indicative of the state of conservation of the product [31].

This is accompanied by the evaluation of the state of oxidation of the oil through the determination and classification of the primary (i.e., hydroperoxides) and secondary (i.e., hydroxy, keto, epoxy, epidioxy) forms of oxidation. From the experience gained in carrying out this determination on a wide range of olive and extra virgin oils, it can be seen that the newly produced oils have a content of oxidized fatty acids < 2%, while the oils at the expiration of storage have a content > 4% [32].

Tocopherols, and in particular vitamin E, are natural antioxidants that protect the lipid components of a food or tissue from oxidation by combining with free radicals and stopping the chain reaction by which free radicals multiply. Their activity depends on the food to which they are added, the concentration used, the availability of oxygen and the presence of heavy metals [31].

In order to highlight the effect of the preparation process, parameters of untreated refined olive oil after the two extraction methods (maceration or UAE—50 mL sample, 7 mm sonotrode, 35% amplitude) were compared and summarized in Table 7.

From the results of the analysis carried out to evaluate the possible changes in the characteristics of the oil due to the implemented extraction processes, no relevant changes were observed. The qualitative characteristics of the oil remained constant for the two treatments and similar to the starting olive oil. Therefore, we can note that no auto-oxidation or photo-oxidation processes have occurred such as to significantly increase the content of volatile carbonyl compounds, and the primary and secondary forms of oxidation of fatty acids. Even the vitamin E content underwent a minimal variation in the oils subjected to treatment, compared to the original sample; the relative oxidized forms showed no significant variation. These results on the negligible effects of ultrasound on the composition of vegetable oils were in agreement with data on almond oil [35] and olive oil [36], as well as on the extraction of bioactive compounds [37].

#### DSC Analysis of Olive Oil

It is known that heat treatments of olive oil can cause chemical and stereochemicals changes on the hydrocarbon chains of fatty acids of triglycerides [38]. These changes are reflected in various physical properties, such as the energy involved in melting and crystallization. 

DSC analysis of olive oil is generally carried out upon cooling a melted state to achieve high reproducibility [39]. As shown in Figure 3, cooled samples undergo to two exothermic events around −10 °C and −40 °C. The former can be reasonably attributed to the phase transition of an oil fraction containing mainly saturated fatty acids, such as palmitic (C 16:0) and stearic (C 18:0) acid [40]. The latter peak at about −40 °C was attributed to crystallization of low melting highly unsaturated oil fraction [40]. 

Both maceration and sonication caused a shift on T_onset_ and a decrease in enthalpy (∆H) with respect to the untreated oil (Table 8). These variations could reflect a modification in composition. Indeed, similar results were obtained in DSC experiments aimed to determine the oxidative stability of olive oil [40] since oxidation can induce depletion of triglycerides, the increase of the free fatty acids content, and/or the increase of viscosity [41]. Moreover, the ability of the crystallizing molecules to come in contact and align to form a crystal detectable under the DSC experimental conditions may have been hindered by the presence of charged and more disordered molecules (increase of polar compounds and low MW compounds) and by the increased viscosity of the oxidized olive oil [42].

Upon heating, the exothermic peak at about −20 °C (Figure 3) could be attributed to the crystallization of the triacylglycerol fraction which did not solidify under the cooling scan used in the current study and/or to the rearrangement of a portion of the crystals formed during cooling [40].

The followed endothermic peak could be reasonably due to the highly unsaturated fraction followed by that of the more saturated one (Figure 3) [40]. Independently of the treatment, modifications of the DSC trace occurred in agreement with literature data [43]: both maceration and sonication caused a massive variation in the T_onset_ and ΔH of the event at about −10 °C (Table 8). Hence, the olive oil composition underwent slight modifications during the treatments. 

## 4. Conclusions

UAE is an efficient and fast extraction technique, and it is here demonstrated to be a successful alternative method in preparing Cannabis extracts for medicinal purposes. After the optimization of all experimental parameters, UAE provides comparable results to maceration, conventionally used by pharmacist in preparing these Cannabis extracts [44]. The same extraction yield was obtained with the advantage of a relevant reduction of the extraction process duration. Moreover, the exposure of the olive oil to cavitation waves during preparation does not affect the qualitative composition of the starting oil. In other words, the organoleptic properties of the oil can be maintained, and the possible alteration of the oily solution cannot be attributed to the UAE. To guarantee this aspect, the storage condition of the olive oil before preparation should be critically considered.

## Figures and Tables

**Figure 1 pharmaceutics-14-02718-f001:**
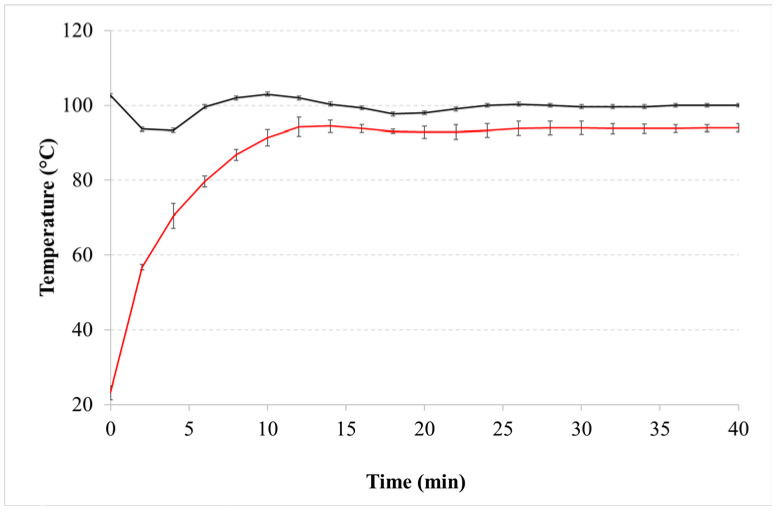
Temperature variation of olive oil (red line) and silicone oil (black line) over time during maceration.

**Figure 2 pharmaceutics-14-02718-f002:**
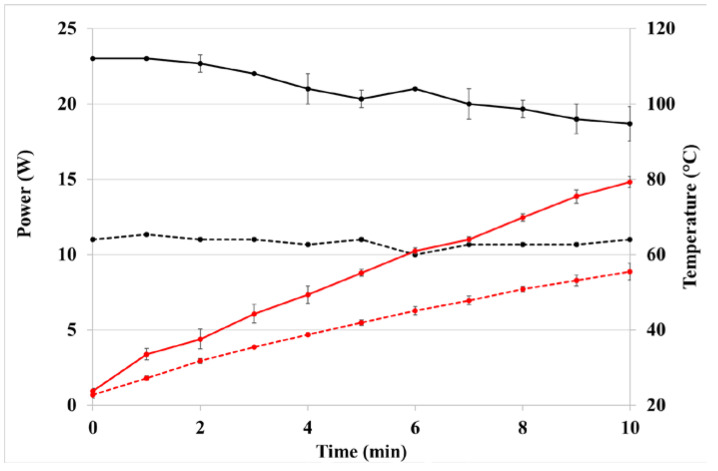
Variation of power (black line) and oil temperature (red line) upon UAE in 50 mL of olive oil using a 2 mm (solid line) or 7 mm (dashed line) probe.

**Figure 3 pharmaceutics-14-02718-f003:**
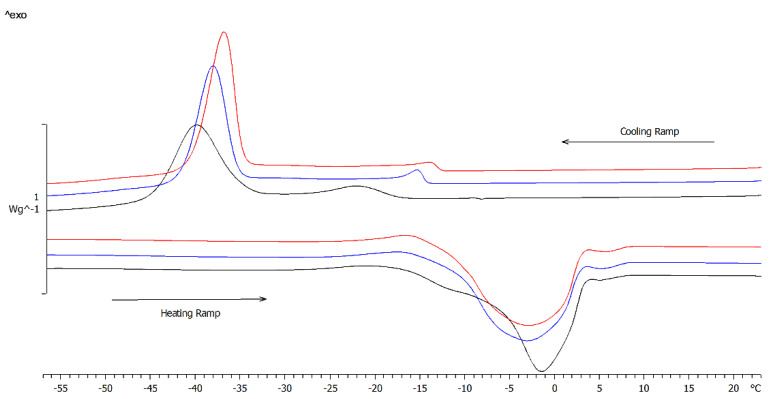
DSC profiles of refined olive oil as raw material (black line) and after maceration for 40 min (red line) or UAE for 10 min using a 7 mm probe (blue line).

**Table 1 pharmaceutics-14-02718-t001:** Experimental conditions [probe diameter (D), amplitude (A) and time] used in ultrasound assisted extraction of cannabinoids from FM2, decarboxylated FM2 (dFM2) and Bedrocan^®^ (dB) at 115 °C in 20 or 50 mL olive oil.

Sample	FM2 (g)	dFM2 (g)	dB (g)	V (mL)	D (mm)	A (%)	Time (min)
1	2	-	-	20	2	60	20
2	-	2	-	20	2	60	10
3	-	2	-	20	2	60	20
4	-	2	-	20	2	60	30
5	-	5	-	50	2	60	10
6	-	5	-	50	7	30	10
7	-	-	5	50	7	30	10
8	-	-	5	50	7	35	10

**Table 2 pharmaceutics-14-02718-t002:** Contents of the main cannabinoids in Cannabis FM2 in the starting material and after decarboxylation at different temperatures, namely 115 and 125 °C. Results are expressed as mean (%, w/w) ± standard deviation (CV%).

Component	FM2 *	Decarboxylation Temperature	
		115 °C **	125 °C ***
CBD	1.17 ± 0.08 (7.07)	5.26 ± 0.34 (6.38)	5.41 ± 0.19 (3.43)
CBDA	6.01 ± 0.38 (6.35)	0.64 ± 0.04 (5.46)	0.22 ± 0.01 (4.40)
CBN	0.23 ± 0.02 (6.96)	0.56 ± 0.04 (6.23)	0.51 ± 0.02 (3.69)
Δ9-THC	0.83 ± 0.07 (7.96)	2.26 ± 0.15 (6.41)	2.34 ± 0.09 (3.66)
Δ9-THCA	2.41 ± 0.16 (6.52)	0.02 ± 0.00 (0)	-
Total CBD	6.44 ± 0.41 (6.38)	5.82 ± 0.37 (6.29)	5.60 ± 0.19 (3.48)
Total THC	2.94 ± 0.20 (6.78)	2.28 ± 0.15 (6.36)	2.34 ± 0.09 (3.66)

Note: * n = 12; ** n = 3; *** n = 4.

**Table 3 pharmaceutics-14-02718-t003:** Cannabinoids content in samples prepared using the maceration method (n = 3), expressed as % w/w (CV%): M-FM2, prepared without decarboxylation; M-FM2_115, prepared with decarboxylated FM2 at 115 °C; M-FM2_125, prepared with decarboxylated FM2 at 125 °C.

Component	No Preheated	Decarboxylated FM2	
	M-FM2 *	M-FM2_115 *	M-FM2_125 **
CBD	0.12 ± 0.01 (4.13)	0.53 ± 0.04 (7.88)	0.70 ± 0.02 (2.86)
CBDA	0.62 ± 0.05 (8.08)	0.07 ± 0.01 (0.00)	-
CBN	0.02 ± 0.01 (24.74)	0.04 ± 0.01 (13.32)	0.02 ± 0.03 (173.21)
Δ9-THC	0.10 ± 0.01 (10.47)	0.26 ± 0.02 (6.66)	0.31 ± 0.01 (1.88)
Δ9-THCA	0.25 ± 0.02 (9.26)	-	-
Total CBD	0.67 ± 0.05 (7.33)	0.59 ± 0.04 (7.57)	0.70 ± 0.02 (2.86)
Total THC	0.32 ± 0.03 (9.29)	0.26 ± 0.02 (6.66)	0.31 ± 0.01 (1.88)

Note: * 20 mL sample; ** 50 mL sample.

**Table 4 pharmaceutics-14-02718-t004:** Cannabinoid content in oily samples prepared by ultrasonic assisted extraction using no heat treated FM2 (t = 20 min) or FM2 decarboxylated at 115 °C. Aliquots of 20 mL sample were sonicated (amplitude 60%) for different time-period (n = 3, expressed as % w/w (CV%).

Component	No Preheated FM2	FM2 Decarboxylated @115 °C		
	20 min	30 min	20 min	10 min
CBD	0.17 ± 0.01 (5.88)	0.63 ± 0.04 (6.04)	0.62 ± 0.08 (12.85)	0.62 ± 0.09 (14.20)
CBDA	0.67 ± 0.03 (4.33)	0.02 ± 0.01 (24.74)	0.03 ± 0.02 (66.67)	0.04 ± 0.03 (72.22)
CBN	0.03 ± 0.00 (0)	0.06 ± 0.00 (0)	0.06 ± 0.01 (13.61)	0.06 ± 0.01 (11.79)
Δ9-THC	0.11 ± 0.01 (5.09)	0.26 ± 0.02 (6.66)	0.26 ± 0.03 (11.32)	0.26 ± 0.02 (8.99)
Δ9-THCA	0.24 ± 0.02 (6.45)	-	-	-
Total CBD	0.75 ± 0.03 (3.61)	0.65 ± 0.04 (6.63)	0.64 ± 0.06 (9.93)	0.66 ± 0.08 (11.66)
Total THC	0.32 ± 0.01 (2.54)	0.26 ± 0.02 (6.66)	0.26 ± 0.03 (11.32)	0.26 ± 0.02 (8.99)

**Table 5 pharmaceutics-14-02718-t005:** Cannabinoids content expressed as % w/w (CV%) in samples prepared with decarboxylated FM2 by ultrasonic extraction for 10 min, using 2 mm (amplitude 60%) or 7 mm (amplitude 30%) sonotrode (50 mL sample; n = 3).

Component	FM2 Decarboxylated @115 °C	
	2 mm Sonotrode	7 mm Sonotrode
CBD	0.49 ± 0.02 (4.22)	0.54 ± 0.02 (2.86)
CBDA	0.08 ± 0.01 (7.53)	0.04 ± 0.01 (3.27)
CBN	0.05 ± 0.00 (0)	0.06 ± 0.00 (0)
Δ9-THC	0.23 ± 0.02 (6.74)	0.19 ± 0.004 (1.98)
Δ9-THCA	-	-
Total CBD	0.56 ± 0.03 (4.60)	0.58 ± 0.01 (2.13)
Total THC	0.23 ± 0.02 (6.74)	0.19 ± 0.004 (1.19)

**Table 6 pharmaceutics-14-02718-t006:** Cannabinoids content expressed as % w/w (CV%) in samples prepared with decarboxylated Bedrocan after maceration and by UAE for 10 min, using 7 mm sonotrode (50 mL sample).

Component	Decarboxylated Bedrocan @115 °C		
	M-BEDROCAN *	US-BEDROCAN-30% *	US-BEDROCAN-35% **
Δ9-THC	1.78 ± 0.17 (9.62)	1.75 ± 0.05 (2.81)	1.94 ± 0.05 (2.74)
Δ9-THCA	0.05 ± 0.03 (51.32)	0.04 ± 0.03 (71.54)	0.04 ± 0.04 (102.60)
Total THC	1.83 ± 0.17 (9.34)	1.79 ± 0.05 (2.75)	1.98 ± 0.01 (0.60)

Note: * n = 3; ** n = 2.

**Table 7 pharmaceutics-14-02718-t007:** Olive oil characterization before and after treatment, maceration or UAE.

		Before Treatment	Maceration			UAE		
			S1	S2	S3	S1	S2	S3
CVCs content (mg/kg)	Hexanal	8	12	12	12	12	12	12
	Nonanal	3	6	6	7	6	6	6
	Total CVCs	11	18	18	19	18	18	18
Oxidized fatty acids content (% w/w)	Primary oxidized forms	0.49	0.57	0.54	0.53	0.57	0.60	0.53
	Secondary oxidized forms	0.90	1.29	1.29	1.45	1.47	1.43	1.42
	Total oxidized FA	1.39	1.86	1.83	1.98	2.04	2.03	1.95
Toc and oxidized Toc (mg/kg)	δ-Toc	9	2	2	<1	<1	<1	<1
	β+γ-Toc	30	38	38	33	36	46	35
	α-Toc	174	157	160	153	146	136	134
	Total Toc	213	197	200	186	182	182	169
	Epoxy α-Toc	1	2	2	1	1	1	1
	TocopherilQ	2	2	2	2	2	2	2
	Total oxidized Toc	3	4	4	3	3	3	3

Note: CVCs: Carbonylic Volatile Compounds; Toc: tocopherols.

**Table 8 pharmaceutics-14-02718-t008:** DSC data on the thermograms of olive oil upon cooling and heating. The term “P1” and “P2” indicates the temperature corresponding to peak temperature of the major and the minimum exothermic event, respectively. “T_onset_” is the onset temperature of the physical event (transition) and “ΔH” is the enthalpy.

Thermal event	Untreated		Macerated		Sonicated	
	Cooling	Heating	Cooling	Heating	Cooling	Heating
P1 (°C)	−39.90 ± 0.05	−1.39 ± 0.11	−36.85 ± 0.08	−2.39 ± 0.50	−37.80 ± 0.25	−2.69 ± 0.57
T_onset_ P1 (°C)	−34.79 ± 0.10	−6.79 ± 0.09	−34.82 ± 0.05	−10.89 ± 0.10	−35.31 ± 0.16	−10.81 ± 0.18
ΔH (J/g) P1	34.66 ± 0.48	−48.03 ± 0.66	33.81 ± 0.31	−51.82 ±1.71	32.09 ± 0.44	−49.92 ±1.85
P2 (°C)	−22.12 ± 0.49	5.43 ± 0.13	−13.95 ± 0.04	6.17 ± 0.21	−15.10 ± 0.21	5.85 ± 0.35
T_onset_ P2 (°C)	−17.84 ± 0.65	4.39 ± 0.14	−12.65 ± 0.04	4.36 ± 0.42	−13.97 ± 0.21	4.03 ± 0.45
ΔH (J/g) P2	4.13 ± 0.10	−0.19 ± 0.05	1.44 ± 0.08	−0.42 ± 0.13	2.22 ± 0.14	−0.52 ± 0.17

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
