# Peer review of "Ultrasound-Assisted Extraction of Cannabinoids from Cannabis Sativa for Medicinal Purpose"

_pharmaceutics, 2022, doi:10.3390/pharmaceutics14122718_

Round 1

Reviewer 1 Report

The manuscript is interesting, but the authors must improve the statistical analysis.  The table 1,2,3,4 and 5. The authors must use between the treatments mean comparative technique as Tukey test. 

Author Response

The Authors would thank for the suggestions. Tables and Figures were edited, according to your suggestions. Moreover, the statistical analysis of data was carried out.

Reviewer 2 Report

The manuscript entitled Ultrasound-assisted extraction of cannabinoids from Cannabis sativa for medicinal purpose is well prepared, easy and interesting to read. The authors should highlight the novelty of the research in abstract. Correct some typing errors and unify the writing of values with units (example: line 221 or 307: 5g to 5 g etc.) The methods are well described, and the results are satisfactorily presented. But some questions arise here:

- At decarboxylation process small difference in temperatures was compared, why 115 and 125 °C and not higher, where the time of process duration can be shortened?

- As maceration of olive oil was carried out at 100°C, are the authors checked the oil quality after thermal processing (other than tocopherols and its oxidized components) and after UAE?

Author Response

The Authors would thank for the suggestions. Below is reported our replay. 

The manuscript entitled Ultrasound-assisted extraction of cannabinoids from Cannabis sativa for medicinal purpose is well prepared, easy and interesting to read. The authors should highlight the novelty of the research in abstract.  

[A2]: The abstract and Introduction was reworded, keeping in consideration the comments’ of all reviewers. 

Correct some typing errors and unify the writing of values with units (example: line 221 or 307: 5g to 5 g etc.). 

[A2]: Typos were corrected. 

The methods are well described, and the results are satisfactorily presented. But some questions arise here:  

(i) At decarboxylation process small difference in temperatures was compared, why 115 and 125 °C and not higher, where the time of process duration can be shortened? 

[A2]: Higher temperatures over 125° were not taken into consideration due to consequent relevant degradation effect on cannabinoids measured as conversion into CBN (Casiraghi et al., 2018). The use of so high temperatures is not recommended. Process duration was not considered; probably it could not be that far away from 40 min.  

(ii) As maceration of olive oil was carried out at 100°C, are the authors checked the oil quality after thermal processing (other than tocopherols and its oxidized components) and after UAE? 

[A2]: In Section 3.6 the effect of the temperature on olive oil in terms of primary and secondary oxidation and oxidative cleavage of fatty acid, both after maceration and UAE treatment, was evaluated. DSC analysis was also performed. 

Reviewer 3 Report

This research shows a large and comprehensive set of new data, and in view of the subject and research topic, it is also of great scientific interest. However, the authors should present more clearly and systematically the results obtained. The paper contains a lot of unnecessary divisions of the text into subsections, which makes the sense lost and it is difficult to follow the results. The results need to be presented and explained more clearly, and I send suggestions for improving the work below.

I suggest the editors to accept the paper after thorough revision.

ABSTRACT - I suggest to avoid abbreviations in the abstract, before further explanations follow, e.g. FM2 and Bedrocan, these trademarks of cannabis can be explained briefly.

INTORDUCTION- I suggest shortening them. For example, lines 94-99 (short description of the experiment) in the introduction are not necessary, only the aim of the study can be stated 2.1. MATERIALS - lines 104-108 this part is rather short and it is necessary to give more information about the materials used. For example for bedrocan the percentage of THC, the amount of material used, for olive oil some additional data from the declaration (e.g. the percentage of unsaturated and saturated fatty acids), of course if available.

2.3.1 Effects of preheating - line 127- "at 115 or 125 °C" - please indicate the exact temperature at which the samples were preheated 2.3.2 Maceration, line 130 - "from finely ground" - since particle size is a very important factor in the extraction process, please indicate the average particle size of the ground sample.

Line 134 - "Preparation was also performed with 5 g of Cannabis and 50 mL of olive oil" - it is not clear for what purpose and with what aim these selected amounts of cannabis and olive oil were used.

2.3.3 Ultrasound assisted extraction, line 138, I suggest that instead of calling the device US homogenizer (it is clear to me that this is the main purpose of this device), you write that the US system with probe is used.

Line 140-141, "2mm sonotrode for samples from 2 to 50 mL" and "7mm sonotrode for samples from 20 to 500 mL" - from the previous text of the description, but also from this part, the setup of the experiment is not clearly explained and it is necessary to refine this part, i.e. clearly explain what factors are varied, the mass of the sample (which), the volume of the solvent (olive oil), the diameter of the probe. In addition, the description of the instruments lacks the nominal power of the instrument. Considering the explanations in the text lines 142-147, I suggest the authors to create a table in which they clearly present the experiment.

Line 161 - avoid unnecessary chapters, e.g. chapter 2.4. is unnecessary, also 2.3. can be coupled with subchapters (2.3. and 2.3.1.; and 2.4. with 2.4.1., 2.4.2, 2.4.3.

2.5 Statistical analysis - it is necessary to obtain information about what statistical software was used.

3. Results and discussion

Lines 220-228 - this part of the text is redundant in the Results and Discussion section as it explains the materials and methods used. Only the results and discussion should be clearly presented in this section.

Same comment for the subsections, all statements like 3.1.1. etc. are unnecessary and I suggest deleting them

Lines 237-239 - same as the previous comments, this part is related to the materials and methods and should be listed in the Results and Discussion chapter.

Suggestion for title of 3.2. chapter - Cannabinoid content in samples prepared by maceration method (instead of "Preparation of FM2 samples according to maceration method" - this kind of title is more convincing for the methods part).

The same suggestion applies to the title 3.3. -  Cannabinoid content in oily samples prepared by ultrasound-assisted extraction.

Line 295-296 "UAE can be carried out using two types of ultrasound equipment: (a) an ultrasonic water bath, or (b)..." - please delete this part, it makes more sense to use it in the introduction part.

Line 297-297 - sentence ". According to results reported in literature, total compounds obtained by UAE bath and UAE sonotrode did not show significant differences" - first, to which compounds does this sentence refer, second, which specific references? Considering that most research and available literature references say exactly the opposite in terms of extraction of bioactive compounds (and other specific compounds, including oily ones), namely that the probe system is significantly more efficient than the bath.

Line 304- "In this work, the sonotrode was first used." - please delete this sentence.

Line 304-306 - delete this part - the same comment as the previous ones, it refers to the methods.

Line 315 - "power of 10 W" - please double check that the power of the US instrument was 10 W, because usually instruments with lower power have a nominal power of at least 100 W.

Lines 313-315 - all of this data should be listed in Materials and Methods, not here.

Line 363 - "Conditions previoulsy fixed for FM2 were used" - delete the sentence.

Conclusion - should be rewritten to show the goal of the study, i.e., highlight the benefits of UAE extraction, show the UAE conditions that were best.

Author Response

The Authors would thank for the suggestions. Below is reported our replay. 

This research shows a large and comprehensive set of new data, and in view of the subject and research topic, it is also of great scientific interest. However, the authors should present more clearly and systematically the results obtained. The paper contains a lot of unnecessary divisions of the text into subsections, which makes the sense lost and it is difficult to follow the results. The results need to be presented and explained more clearly, and I send suggestions for improving the work below. 

I suggest the editors to accept the paper after thorough revision. 

ABSTRACT - I suggest avoiding abbreviations in the abstract, before further explanations follow, e.g. FM2 and Bedrocan, these trademarks of cannabis can be explained briefly. 

[A3]: The sentence was reworded, according to your suggestion. 

INTRODUCTION- I suggest shortening them. For example, lines 94-99 (short description of the experiment) in the introduction are not necessary, only the aim of the study can be stated 2.1.  

[A3]: The introduction was shortened. 

MATERIALS - lines 104-108 this part is rather short and it is necessary to give more information about the materials used. For example, for Bedrocan the percentage of THC, the amount of material used, for olive oil some additional data from the declaration (e.g. the percentage of unsaturated and saturated fatty acids), of course if available. 

[A3]: All required additional information on raw materials was added. Details of the preparation are reported in Section 2.3.  

2.3.1 Effects of preheating 

line 127- "at 115 or 125 °C" - please indicate the exact temperature at which the samples were preheated 

2.3.2 Maceration, line 130 - "from finely ground" - since particle size is a very important factor in the extraction process, please indicate the average particle size of the ground sample. 

[A3]: Description concerning temperatures used for decarboxylation was reworded.  

Herbal derived products, i.e. Cannabis, are generally grounded to remove wooden parts as their mass would affect the weight. In other words, one would not weigh just wooden debris not containing active substance. 

Line 134 - "Preparation was also performed with 5 g of Cannabis and 50 mL of olive oil" - it is not clear for what purpose and with what aim these selected amounts of cannabis and olive oil were used. 

[A3]: For this type of preparation, the most used prescription is 5g in 50mL, that is the whole Cannabis container.  This explanation was added in the manuscript. 

2.3.3 Ultrasound assisted extraction, line 138, I suggest that instead of calling the device US homogenizer (it is clear to me that this is the main purpose of this device), you write that the US system with probe is used. 

[A3]: the Authors called this device “homogenizer”, as reported on the website of the dealer: https://www.hielscher.com/up200st-powerful-ultrasonic-lab-homogenizer.htm 

The text was modified according to your suggestion.  

Line 140-141, "2mm sonotrode for samples from 2 to 50 mL" and "7mm sonotrode for samples from 20 to 500 mL" - from the previous text of the description, but also from this part, the setup of the experiment is not clearly explained and it is necessary to refine this part, i.e. clearly explain what factors are varied, the mass of the sample (which), the volume of the solvent (olive oil), the diameter of the probe. In addition, the description of the instruments lacks the nominal power of the instrument. Considering the explanations in the text lines 142-147, I suggest the authors to create a table in which they clearly present the experiment. 

[A3]: The concentration of Cannabis in olive oil remained constant at 1%. The volume of olive oil was considered at two levels, namely 20 mL and 50 mL. The rational is related to the different diameters of the sonotrode required to carry out the extraction. The ranges are defined by the constructor of the ultrasonic device. Details of the system were added. The text was modified accordingly and a Table to describe the experiment was added.  

Line 161 - avoid unnecessary chapters, e.g. chapter 2.4. is unnecessary, also 2.3. can be coupled with subchapters 2.3. and 2.3.1.; and 2.4. with 2.4.1., 2.4.2, 2.4.3. 

[A3]: Unnecessary chapters were removed. 

2.5 Statistical analysis - it is necessary to obtain information about what statistical software was used. 

[A3]: According to your suggestion, statistical evaluation was improved.  

RESULTS AND DISCUSSION 

Lines 220-228 - this part of the text is redundant in the Results and Discussion section as it explains the materials and methods used. Only the results and discussion should be clearly presented in this section. Same comment for the subsections, all statements like 3.1.1. etc. are unnecessary and I suggest deleting them. Lines 237-239 - same as the previous comments, this part is related to the materials and methods and should be listed in the Results and Discussion chapter. 

[A3]: Redundant part of the text and unnecessary subsections were removed. 

Suggestion for title of 3.2. chapter - Cannabinoid content in samples prepared by maceration method (instead of "Preparation of FM2 samples according to maceration method" - this kind of title is more convincing for the methods part). The same suggestion applies to the title 3.3. -  Cannabinoid content in oily samples prepared by ultrasound-assisted extraction. 

[A3]: The titles of 3.2 and 3.3 chapters were modified. 

Line 295-296 "UAE can be carried out using two types of ultrasound equipment: (a) an ultrasonic water bath, or (b)..." - please delete this part, it makes more sense to use it in the introduction part. 

[A3]: According to your suggestion, this part is moved in the Introduction.  

Line 297-297 - sentence "According to results reported in literature, total compounds obtained by UAE bath and UAE sonotrode did not show significant differences" - first, to which compounds does this sentence refer, second, which specific references? Considering that most research and available literature references say exactly the opposite in terms of extraction of bioactive compounds (and other specific compounds, including oily ones), namely that the probe system is significantly more efficient than the bath. 

[A3]: According to your suggestion, this part was improved.  

Line 304- "In this work, the sonotrode was first used." - please delete this sentence. Line 304-306 - delete this part - the same comment as the previous ones, it refers to the methods. Line 363 - "Conditions previoulsy fixed for FM2 were used" - delete the sentence. Lines 313-315 - all of this data should be listed in Materials and Methods, not here. 

[A3]: This paragraph was reworded. 

Line 315 - "power of 10 W" - please double check that the power of the US instrument was 10 W, because usually instruments with lower power have a nominal power of at least 100 W. 

[A3]: Nominal power of the US instrument was added.  

CONCLUSION - should be rewritten to show the goal of the study, i.e., highlight the benefits of UAE extraction, show the UAE conditions that were best. 

[A3]: Conclusion was rewritten. 

Reviewer 4 Report

The work is very interesting and the results can be important to improve the extraction processes, especially in saving time. However, nothing is discussed about cost. Maceration takes longer but in economic terms it can be more viable at an industrial level.

In Figures and Tables the uniform nomenclature of "Figure 1. Title" or "Table 1. Title." Review Table 7 and all the Figures...

Author Response

The Authors would thank for the suggestions. Below our replay is reported. 

The work is very interesting and the results can be important to improve the extraction processes, especially in saving time. However, nothing is discussed about cost. Maceration takes longer, but in economic terms it can be more viable at an industrial level. 

In Figures and Tables the uniform nomenclature of "Figure 1. Title" or "Table 1. Title." Review Table 7 and all the Figures... 

[A4]: All figures were re-edited and nomenclature was made uniform. In the work we discussed the use of US system in pharmacy as an alternative to maceration; for this reason, cost was less relevant. 

Round 2

Reviewer 1 Report

The corrections were done.

Reviewer 3 Report

The authors have taken all suggestions into consideration and the paper is now of satisfactory quality and suitable for publication.